# New Prospects of Waste Involvement in Marine Fuel Oil: Evolution of Composition and Requirements for Fuel with Sulfur Content up to 0.5%

Dmitriy V. Nelyubov [1], Marat I. Fakhrutdinov [1], Alena A. Sarkisyan [1], Evgeniy A. Sharin [2], Mikhail A. Ershov [3,4,*], Ulyana A. Makhova [3], Alisa E. Makhmudova [3], Nikita A. Klimov [3], Marina Y. Rogova [3], Vsevolod D. Savelenko [3], Vladimir M. Kapustin [3,4], Marina M. Lobashova [3] and Ekaterina O. Tikhomirova [3]

1   The 25th State Research Institute of Chimmotology of the Ministry of Defense of Russia, 121467 Moscow, Russia
2   A.N. Severtsov Institute of Ecology and Evolution of the Russian Academy of Sciences, 119071 Moscow, Russia
3   Department of Oil Refining Technology, Gubkin Russian State University of Oil and Gas (National Research University), 119991 Moscow, Russia
4   Academy of Engineering, Peoples' Friendship University of Russia (RUDN University), 117546 Moscow, Russia
*   Correspondence: ershovma@ntwc.ru

**Abstract:** Research was carried out on the possibility of involving oil refining wastes and petrochemical by-products in marine fuel oil. It was shown that the properties of the studied products (VAT distillation residue of butyl alcohols, heavy pyrolysis tar, desalted phenol production tar, waste motor oil mixture) mainly differ from primary and secondary oil refining products used in this fuel with increased toxicity (hazard classes 2 and 3). A clear disadvantage of waste motor oils is an increased content of metals, particularly zinc, calcium and phosphorus, which leads to high ash content. Recommended concentrations for introducing components into marine fuels are given. The influences of the composition and sulfur content on operational properties and quality indexes of VLSFO were also studied. It is shown that the use of products of deep hydrotreatment of vacuum-distillate fractions of oil processing can worsen its protective (anticorrosive) properties and colloidal stability; therefore, a reduction of sulfur content below 0.1% in this fuel is inexpedient without the use of additives. The requirements for VLSFO quality indicators have been developed. Application of VLSFO corresponding to the developed requirements will provide an increase in performance of ship power plants and the stability of VLSFO quality, which will contribute to cost reduction of ship owners when using it.

**Keywords:** marine fuel; very low sulfur fuel oil; waste lubrication oil; marine fuel oil; phenol tar; heavy pyrolysis tar

## 1. Introduction

Marine fuel quality control has undergone significant changes in recent years since 01.01.2020 and the introduction of restrictions on the sulfur content (Max 0.5 % wt.) in the fuel used on ships not equipped with emission treatment systems by the MARPOL 73/78. This convention has had a serious impact on the oil refining and shipping industry and the oil and oil products market in general, on the balance of the supply and demand prices. From 12 to 16 December 2022, the 79th session of the Marine Environment Protection Committee (MEPC 79) was held in London, following which a number of resolutions were adopted that amend MARPOL and other regulations and prescriptions [1]. The most important of these was the establishment of a new emission control area (ECA). This new area is the Mediterranean Sea, bounded by the Strait of Gibraltar, the Dardanelles and the

Suez Canal. According to resolution MEPC.361 (79), amendments to MARPOL Annex VI will come into force on 1 May 2024, and the restriction on the sulfur content in fuel itself is not more than 0.1 % wt. will come into effect on 1 May 2025.

At the moment the content of sulfur, nitrogen oxides and particulate matter is regulated, but carbon dioxide emissions are not standardized [2]. Since 2021, there have been five proposals to develop carbon pricing methodologies for ships [3]. The schemes have been proposed by the Marshall and Solomon Islands, Japan, Argentina, Norway, Austria and others. Each scheme has its own characteristics in terms of emissions coverage (whole life cycle or use only) and greenhouse gases considered (carbon dioxide only or all of them). However, none of the methods are currently accepted by IMO; moreover, most of them need significant improvement. The issue of market-based carbon regulation of shipping was already raised in the IMO more than 10 years ago, when 11 proposals were submitted for discussion in 2010. However, the report on these initiatives issued by the organization and the meetings that followed did not lead to any decision and the discussion of this issue was discontinued in 2013. Thus, it is expected that soon the necessary regulations will be developed and implemented in the industry, which means the prospect of research into fuels with a lower carbon footprint is soon to be flourishing.

To meet the new requirements of IMO 2020, it is possible to use the following approaches [4]:

1. Installation of scrubbers or other exhaust gas cleaning systems on bunkered vessels and use of heavy fuel (HFO). Installation of scrubbers requires capital expenditures, as well as the need for repair and maintenance of the used devices [5]. Although analytical calculations show a fairly short payback period, the use of scrubbers is limited due to the small number of manufacturers of purification systems [6]. In addition, the use of scrubbers, although it reduces the amount of emissions into the air, pollutes the marine environment with polycyclic hydrocarbons that adversely affect marine organisms [7].

2. The use of LNG (liquefied natural gas) is more preferable from the environmental point of view: there is a significant reduction in emissions of sulfur oxides and nitrogen oxides, as well as the carbon footprint reduction, albeit not a nullification but significant costs are required for the reconstruction of fuel systems and the construction of a new type of ship [8]. It should also be noted that a number of studies recognize that the environmental benefit of LNG is overestimated due to possible methane leaks [9], which negates the positive effect of $CO_2$ nitrogen and sulfur oxides removal. Problem solving can be associated with the use of biogas by the calculation of predicted risk [10]. Uncontrolled emissions range from 0.1 to 3% of total production, depending on purification technologies [11].

A key fact in the use of alternative fuels is the way the feedstock is obtained. For example, producing methanol from natural gas (the most common method) emits an average of 110 g $CO_2$ eq./MJ. At the same time, coal may be used as a feedstock, which is currently done only in China, and methanol has a significantly larger carbon footprint about of 300 g $CO_2$ eq./MJ. The production of methanol from renewable sources (biomethane, biomass, municipal solid waste) using renewable energy shows significantly less emissions, averaging 10–40 g $CO_2$ eq./Mg [12]. The use of ammonia as marine fuel brings a significant advantage, since the $CO_2$ emissions are significantly lower than for methanol, LNG and MGO (marine gas oil). However, a specific need for ammonia use is its trait of removing nitrogen oxides from the exhaust gases [13].

As well as the use of alternative fuels in engines, it is also possible to use fuel cells as the power plants that convert chemical energy from fuel into electricity [14]. Green and blue hydrogen and ammonia [15] are seen as promising fuel resources in studies. The disadvantage of such fuels is the high cost of production.

3. The electrification of ships is also a promising direction of powering the ships due to the noticeable environmental benefits in terms of reducing emissions of greenhouse gases, sulfur oxides, nitrogen and particulate matter [16]. This perspective is underlined by a study of the behavior of shipowners in Norway. Of all alternatives, electricity [17] is given the greatest preference. In this case, as with electric vehicles, the most difficult part is

providing ships with green electricity. In the case of a sufficient share of renewable energy in total energy consumption—for example, 46% in Croatia—electrification can significantly reduce life cycle emissions compared to other alternatives [18].

4.  Another, less expensive use from the point of view of re-equipment of the ship engine design is the use of biofuels based on renewable vegetable raw materials [19]. At the same time, the use of biocomponents—for example, biodiesel mixed with petroleum fuel—makes it possible to improve its performance properties, such as viscosity, lubricity and low-temperature properties [20]. Distribution of biofuels as energy carriers for marine transport is hindered by the limited resources for their production (oilseed crops), which often compete with the more demanded crops used in the food industry in connection with which non-feed and non-food raw materials—various agricultural, forest and other wastes should be used as raw materials [21]. Waste recycling processes are more complicated in terms of technology, which leads to an increase in the cost of the final fuel [22].

5.  Application of low-sulfur marine fuels: VLSFO (very-low-sulfur fuel), ULSFO (ultra-low-sulfur fuel oil) or distillate marine fuel [2]. According to Transport&Environment [23], the share of traditional oil ship fuel (as an energy carrier) in the volume of energy carriers used in the civil fleet is 94%. Liquefied natural gas accounts for the remaining 6%. As of the beginning of 2022, the share of biofuels in the supply of civil shipping is insignificant. However, it is predicted to increase to 2% by 2025. Thus, despite the emerging trends in decarbonization, petroleum fuels (VLSFO, ULSFO) will play a key role until 2040, which makes it relevant to develop new fuel compositions and assess the involvement of oil refining waste.

Marine residual fuels are obtained by mixing heavy petroleum products of both primary and secondary origin and distillate products [24–27]. Low-sulfur products of hydrogenation processes are of particular interest for compounding. These processes include hydrotreating of middle distillates, vacuum distillates and oil residues, carried out under hydrogen pressure up to 3–4 MPa and a degree of hydrocarbon transformation of Max 10%; hydrocracking and hydroconversion of vacuum distillates, heavy gas oils and residues of thermal, catalytic cracking; and coking, which provide a degree of fuel hydrocarbon transformation of over 90%. Figure 1 shows the components which can be used for production of marine fuel VLSFO, and the color shows components which are the most commonly used according to the results of research of actual fuel quality in the key ports of the world [28].

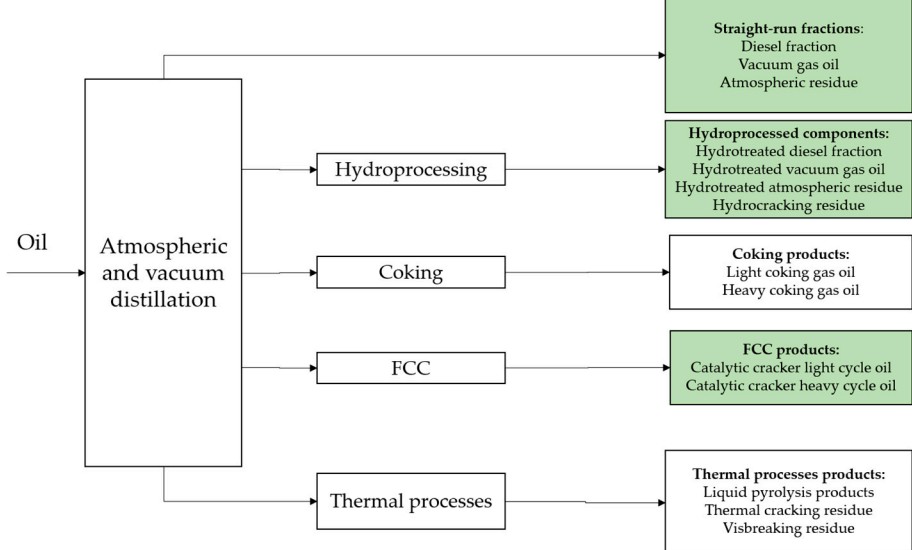

**Figure 1.** VLSFO production scheme.

The most popular VLSFO marine fuel brand is RMG 380, the share of which is about 90% of the total VLSFO quantity. At the same time, "true" RMG 380 fuels, the viscosity of which would exceed the 180 cSt. Limit for the RMG 180 grade, make only about 13%, i.e., the majority of marine fuel producers unify the labeling of heavy VLSFOs under RMG 380 grade.

All potential components, which can be used in marine fuels, have their advantages and disadvantages in terms of their properties and sulfur content (Table 1). The straight-run components and products of thermal processes (coking, visbreaking) limit their use in the composition mainly because of their high sulfur content. At the same time, hydrotreated fractions or residues have high added value; in the case of hydrotreated diesel fraction, the economic feasibility of selling diesel fuel is much higher than ship diesel fuel. In the case of hydrotreated residues and hydrotreated vacuum gas oil (HVGO), there is a demand to use them as a feedstock for other catalytic processes such as catalytic cracking, for example.

**Table 1.** The possibility of involving various oil components in the VLSFO composition using data from [29–34].

| Component | Involvement Limit, % | Limiting Characteristics/Property |
|---|---|---|
| Straight vacuum gas oil | 30 | The sulfur content, which is usually from 1.0 to 3.0%. |
| Hydrotreated vacuum gas oil | 100 | Possibility of use as a feedstock to obtain higher-margin products. |
| Atmospheric residue | 20 | The sulfur content, which is usually from 1.5 to 4.0%. Aggregative stability of the fuel, which also depends on the overall composition of the composition. |
| Hydrotreated atmospheric residue | 95 | High viscosity, which, with a high degree of selection of light fractions at the hydrotreating unit, can exceed 380 cSt. |
| Straight-run diesel fraction | 50 | The sulfur content, which is usually from 0.2 to 2.0%. Aggregative stability of the fuel, which also depends on the overall composition of the composition. |
| Hydrotreated diesel fraction | 50 | The aggregative stability of the fuel, which also depends on the overall composition of the composition. Economic feasibility: the selling price of a diesel fuel is higher than that of a ship. |
| Light coking gas oil | 30 | The sulfur content, which is usually from 0.8 to 2.5%. Oxidative stability of fuel due to high content of olefins. |
| Heavy coking gas oil | 20 | The sulfur content, which is usually from 1.5 to 5.0%. Oxidative stability of the fuel due to the high content of olefins. |
| Catalytic cracker light cycle oil (with preliminary hydrotreatment of feedstock/without it) | 50/30 | The sulfur content, which is usually from 0.2 to 0.5% of the preliminary hydrotreating feedstock and 0.8 to 2.5% without it. High calculated aromaticity index (low flammability). |
| Heavy coking gas oil (with preliminary hydrotreatment of feedstock/without it) | 50/0 | The sulfur content, which is usually from 0.3 to 0.7% with preliminary hydrotreating of feedstock and 3.0 to 5.0% without it. High estimated aromaticity index (low flammability). High concentration of catalyst dust (Al and Si). |
| Visbreaking residue | 15 | The sulfur content, which is usually from 1.8 to 4.5%. Aggregative stability of the fuel: due to the thermal origin of the asphaltene fraction, it is extremely unstable. |
| Liquid pyrolysis products | 20 | Aggregative stability of the fuel: due to the thermal origin of the asphaltene fraction in it, it is extremely unstable. High calculated aromaticity index (low flammability). High water content. |
| Hydrocracking residue | 100 | Possibility of use as a raw material for obtaining higher-margin products. |

Based on the review above, the knowledge gap is indicated as follows:

- Despite a large number of works on the study of alternative fuels, the use of oil refining and petrochemical waste as components of marine fuel is rarely studied.

The high cost of alternative fuels is expected to decrease with the development of technology but, in many respects, limits their implementation at the moment. Thus, the task of this work was to investigate the possibility of using by-products (waste products) of oil and petrochemical processing, used or recommended for use by manufacturing enterprises, as components of marine fuels. The second task of this work was to study the properties of low-sulfur marine fuels in order to develop additional quality requirements for VLSFO to produce high-quality marine fuels.

## 2. Materials and Methods

### 2.1. Materials

Phenol tar of producer № 1; a VAT distillation residue of butyl alcohols (VAT DRBA) and a heavy product of rectification of 2-ethylhexanol (HPRT) of producer № 2; heavy pyrolysis tar of producers № 3, № 4 and № 5; and waste motor oils of producer № 6 were considered as waste of oil refining and petrochemistry in this work. Quality indicators of oil components to study the properties of fuels with different sulfur content are shown in Table 2.

**Table 2.** Quality indicators of oil components.

| Property | Distillate Low-Viscosity Marine Fuel | Medium Vacuum Gas Oil Distillate | Light Vacuum Gas Oil Distillate | Heavy Oil Fuel | Hydrotreated Vacuum Gas Oil | Diesel of VG Hydrotreatment |
|---|---|---|---|---|---|---|
| Kinematic viscosity at 100 °C according to ISO 3104, mm$^2$/s | - | 5.911 | 1.68 | 2.426 | 6.04 | 0.99 |
| Kinematic viscosity at 50 °C according to ISO 3104, mm$^2$/s | - | - | - | - | 26.83 | 1.90 |
| Up to 360 °C distilled ASTM D 1160, % vol. | 95 | - | 93 | - | - | - |
| Flash point in closed cup ISO 2719, °C | 82 | - | 88 | 71 | >150 | 60.5 |
| Density at 15 °C according to ISO 3675, kg/m$^3$ | - | - | - | 858.40 | 894.29 | 856.53 |
| The content of water-soluble acids and alkalis GOST 6307 | absence | absence | absence | absence | absence | absence |
| Sulfur content according to ISO 8754, % wt. | 0.349 | - | - | 0.366 | - | - |
| Carbon residue according to ISO 10370, % | - | 0.070 | - | 0.39 | 0.016 | absence |
| Ash content according to ISO 6245, % | - | - | - | 0.006 | 0.003 | absence |
| Pour point according to ISO 3016, °C | minus 13 | - | - | minus 10 | 20 | - |
| Optical density of sediment solution, units | - | - | - | - | 0.020 | 0.020 |
| Volume of emulsified water, cm$^3$ | - | - | - | - | 3.0 | 0.5 |

### 2.2. Methods

The methods used in this work for the analysis of potential components of marine fuels correspond to the methods of ISO 8217. Besides ISO 8217 methods, additional methods of qualification evaluation are used such as the method of laboratory tests of fuel biostability (GOST 9.023) and the method of corrosion aggressiveness determination according to a standard of organization (SOO).

Determination of biostability is carried out by the contamination of fuel in contact with an aqueous mineral environment by specially selected microorganisms. The used cultures are the fungi *Cladorium resinae* and the bacteria *Pseudomonas aeruginosa*. For testing, the most aggressive strains of pure cultures and enrichment cultures are selected, which give abundant growth on the original sample for comparison after no more than 14 days of cultivation (30 days for the fungus). Cultures of bacteria and fungi in amounts of 0.3 mL are placed into the medium, prepared in advance with a pipette, and 3 mL of the tested fuels

are added there. Tubes with fungus are kept for 21 days at a temperature of $(29 \pm 2)$ °C, and for 7 days with bacteria with continuous shaking at the same temperature. If signs of growth appear before the specified dates, the tests are stopped. At the end of the test, the test tubes are inspected for the absence of biofilms at the medium–fuel interface (without shaking), the presence of sediment, transparency and the absence of pigmentation.

The essence of the method for determining the corrosion aggressiveness is to assess the change in weight of the metal plate after four hours of alternate exposure to air and fuel and water heated to 80 °C, referred to the unit area of the plate and expressed in $g/m^2$. The method is performed on the Pinkiewicz apparatus [35]. For each fuel sample, two steel plates are taken, treated with an emery cloth, washed with an alcohol–toluene mixture (3:2), dried between sheets of filter paper and weighed. A total of 35 cm$^3$ of the fuel to be tested and 50 cm$^3$ of an aqueous solution of NaCl (17 g of salt/1 L of distilled water) are poured into a glass tube. The plates are immersed in the solution and the motor is turned on; with the help of a rocking mechanism, the plates perform reciprocating movements, being at regular intervals in water, fuel and air. After 4 h at a temperature of $(80 \pm 2)$ °C, the plates are removed, washed and treated with inhibited hydrochloric acid for 1 min (0.8–1.0 g of inhibitor with 55 cm$^3$ of concentrated acid and 45 cm$^3$ of distilled water) to remove corrosion products from the surface of the plate. Next, neutralization is carried out in a Na$_2$CO$_3$ solution, washed with a large amount of water, dried and weighed. For each plate taken for testing, the etch constant is determined. For this, the plates, prepared in the same way as for the experiment, are etched in inhibited hydrochloric acid for one minute.

The value of the corrosion index K ($g/m^2$) is calculated with the formula:

$$K = \frac{(W_1 - W_2) - A}{B},$$ (1)

$W_1$—weight before, g;
$W_2$—weight after, g;
$A$—plate constant, g;
$B$—area of all 6 facets of the surface, m$^2$.

The corrosive aggressiveness of the fuel during the 4 h test is calculated as the arithmetic mean of 2 parallel tests. Permissible discrepancies, depending on the range of corrosion values, vary from 0.3 (with a value of up to 2.5) to 1.9 (with a corrosion value of more than 15).

The toxicity of a mixture of compounds is calculated in accordance with GOST 32423-2013 hazard classification of mixed chemical products by effects on the body according to the following formula:

$$ATE_{mix} = \frac{100}{\sum_{i=1}^{n} \frac{C_i}{ATE_i}},$$ (2)

$ATE_i$—the assessment of the acute toxicity of the component $i$;
$C_i$—concentration of the component $I$ of the mixture;
$n$—number of components;
$ATE_{mix}$—the assessment of the acute toxicity of the blend.

The ATE value for calculating the hazard class of the mixture as a whole is calculated from the range of DL$_{50}$ values separately for ingestion and dermal exposure using the table given in the GOST 32423-2013.

## 3. Results

### 3.1. Phenol Tar

Phenol tar is a VAT distillation residue of cumene hydroperoxide decomposition products, a waste product of the acetone and phenol production processes. At the same time about 200 kg of phenol tar is formed per 1 ton of phenol. Quality indicators of desalted phenol tar, presented in [36] (Table 3), indicate that this product can be a component, involved in

VLSFO with sulfur content of Max 0.5 % wt. according to the following criteria: ash content, viscosity and sulfur content. Phenol tar contains a large number of components: phenol, acetophenone, dimethylphenylcarbinol (DMPC), dimer-methylstyrene, para-cumylphenol (PCP), as well as other products and small amounts of salts (mainly $Na_2SO_4$) [37]. The exact composition of phenol tar depends on the specific phenol production technology. At the moment, most phenol tars are used as a fuel in the reaction equipment; however, there are studies on the maximum use of the chemical potential of the tar [38].

**Table 3.** Quality indicators of initial and desalted phenol tar in comparison with the requirements for VLSFO.

| Property | Phenol Tar before Desalting | Phenol Tar after Desalting | RMG-380 According to ISO 8712 |
|---|---|---|---|
| Appearance | Dark viscous liquid | | - |
| Na content, 10–4, % wt. | 133–336 | ≤6 | Max 100 mg/kg |
| Flash point, °C | 105 | 95 | Min 60 |
| Conditional viscosity, °VU at temperature | 7.5 (30 °C) | 4.0 (30 °C) | Kinematic viscosity < 380 (50 °C) |
| Phenol content, % wt. | 8 | 7 | - |
| Sulfur content, % wt. | 0.3 | ≤0.2 | Max 0.5 |
| Carbon residue, % wt. | 5.0 | ≤2.2 | - |
| Ash content, % wt. | 0.22 | ≤0.05 | Max 0.1 |
| Mechanical impurities content, % wt. | 11.4 | absence | - |
| Water content, % wt. | - | absence | Max 0.5 |
| Specific heat of combustion, kJ/kg | 37,254 | 40,740–42,420 | - |

One disadvantage of the proposed component is the presence of corrosive sodium, which causes high-temperature corrosion and the formation of a strong thermal insulating layer on the heating surfaces. Nevertheless, the desalting process can significantly reduce the sodium content to less than 6 ppm. Another more significant disadvantage is also the high content of phenol, which is a highly hazardous substance of hazard class 2, which does not meet the requirements for VLSFO, which is a substance of hazard class 4 (low hazard).

The hazard class of a substance is determined by a number of indicators, including MPC in the air of the working area, average lethal doses when applied to the skin, in the air and so on. Accordingly, fuels containing waste with a high hazard class may need additional studies to select the optimal concentration. With a phenol tar content of 7% phenol, the approximate concentration proposed in this paper is Max 5%, noting that engine test to refine the concentration is needed.

Another way, which requires a separate assessment, is the possibility of improving the quality of phenol tar by removing phenols from its composition. One of these methods, the oldest one, is phenol extraction with an aqueous solution of ammonia with a phenol removal efficiency of up to 97% [39,40]. Thus, an additional purification stage can help to improve the quality of phenol tar for its use in marine fuels.

### 3.2. Butyl Alcohol Products (VAT DRBA and HPRT)

In [41], the possibility of using the VAT distillation residue of butyl alcohols (VAT DRBA) and the heavy rectification product of 2-ethylhexanol (HPRT) as a component of VLSFO was indicated. Quality indicators of VAT DRBA are shown in Table 4. HPRT characterized by higher density and a higher flash point can also be used.

In terms of all the above quality indicators, VAT DRBA and HPRT meet the requirements for VLSFO components. The advantages of this product as a distillate fuel component include low yield point, no sulfur, a high flash point and high contents of monoaromatic compounds, which provide dispersion of asphaltene associates.

**Table 4.** Quality indicators of initial and desalted phenol tar in comparison with the requirements for VLSFO.

| Property | Units | Result | RMG-380 (ISO 8712) |
|---|---|---|---|
| Density at 15 °C | kg/m$^3$ | 917.7 | Max 991 |
| Density at 20 °C | kg/m$^3$ | 913.9 | - |
| Kinematic viscosity at 50 °C | mm$^2$/s | 4.789 | Max 380 |
| Viscosity conditional (Engler) at 50 °C | °Э | 1.38 | - |
| Kinematic viscosity at 100 °C | mm$^2$/s | 1.415 | - |
| Sulfur content | % wt. | <0.0017 | Max 0.5 |
| Flash point in closed cup | °C | 78.0 | Min 60 |
| Water content | % wt. | 0.65 | Max 0.5 |
| Pour point | °C | <42 | Max 30 |
| Total nitrogen | % wt. | 0.004 | - |
| Aniline point | °C | 11.1 | - |
| Carbon residue according to Conradson | % wt. | | - |
| Distillation temperature | | 70 | - |
| Boiling point | °C | 162 | - |
| 5% recovered | °C | 186 | - |
| 10% recovered | °C | 199 | - |
| 20% recovered | °C | 212 | - |
| 30% recovered | °C | 228 | - |
| 40% recovered | °C | 236 | - |
| 50% recovered | °C | 244 | - |
| 60% recovered | °C | 248 | - |
| 70% recovered | °C | 258 | - |
| 80% recovered | °C | 280 | - |
| 90% recovered | °C | 291 | - |
| 95% recovered | °C | 291 | - |
| Final boiling point | °C | 0.0 | - |
| Distillation loss | % vol. | 5.0 | - |
| Recovered at 360 °C | % vol. | 4.0 | - |
| Bromine number | Br$_2$/100 g | | - |
| Content of saturated, aromatic and polar compounds: | | <5 (0.3) | - |
| Saturated hydrocarbons | % wt. | 5.9 | - |
| Aromatic hydrocarbons | % wt. | 90.6 | - |
| polar univalent | % wt. | 3.2 | - |
| polar divalent | % wt. | 0 | - |
| Toluene equivalent | % vol. | 0 | - |
| Xylene equivalent | % vol. | 0 | - |
| Peptization number | | >5 | - |
| Total sediment | % wt. | 0.01 | Max 0.1 |
| Metal content: | | | |
| V | ppm | <1 | Max 350 |
| Na | ppm | 29 | Max 100 |
| Ni | ppm | 3 | - |
| Al | ppm | 20 | Sum– |
| Si | ppm | 12 | Max 60 |
| Fe | ppm | 55 | - |
| Zn | ppm | 2 | Max 15 |
| Ca | ppm | 5 | Max 30 |
| Color | | D 8 | - |
| Asphaltene content | % wt. | 0.24 | - |

The main disadvantage of VAT DRBA and HPRT when used in the composition of VLSFO is an increased hazard class 3 (moderately hazardous substance) instead of 4 (low-hazardous substance) according to the safety data sheet [42]. In this connection, the use of this product in VLSFO is only possible in the mixed component formulation with a recommended concentration of Max 15%.

*3.3. Heavy Pyrolysis Tar*

According to the data of the invention [43] heavy pyrolysis tar treated according to the method by heating to 110 °C and air (nitrogen or flue gases) barbotage to remove water and light fractions can be used as a component of VLSFO. Test results of this product as a component of residual marine fuels presented in [44] showed that sulfur content in different batches meets the requirements of MARPOL 73/78 and varies from 0.02 to 0.11 % wt. (Table 5).

**Table 5.** Quality indicators of heavy pyrolysis tar in comparison with the requirements for VLSFO.

| Property | HPT Brand A | HPT Brand B | RMG-380 по ISO 8712 |
|---|---|---|---|
| Density at 20 °C, кг/м$^3$ | Min 1040 | Min 1030 | Max 991 |
| Na content, ppm | Max 50 | Max 100 | Max 100 |
| Flash point, °C | 105 | 95 | not less than 60 |
| Kinematic viscosity at 50 °C, сСт | Max 25 | Max 40 | Max 380 |
| Sulfur content, % wt. | Not standardized, 0.02–0.11% | | Max 0,5 |
| Carbon residue, % wt. | Max 12 | Max 16 | Max 18 |
| Pour point, °C | Not standardized, varies from minus 24 to minus 42 | | Max 30 |
| Mechanical impurities content % | Max 0.01 | Max 0.01 | - |
| Water content, % | Max 0.3 | Max 0.5 | Max 0.5 |
| Asphaltene content, % | Not standardized, 6.3–10.5 | | - |
| Flash point, °C | Not standardized, 23–56 | | Min 60 |

At the same time, a number of significant drawbacks have been identified:

- High density at 15 °C from 1002.2 kg/m$^3$ to 1058.4 kg/m$^3$ is unacceptable for VLSFO due to its contact with water in the fuel tanks of vessels (ships) and the necessity of water separation from fuel prior to its supply to the power plant. It is worth noting, however, that there are grades of residual fuels with a density requirement of no more than 1010 kg/m$^3$, which, however, is still below the actual quality of heavy pyrolysis tar;
- A low flash point in a closed cup from 23 °C to 56 °C does not provide the level of fire safety established by the international requirements for the safety of navigation, established by the MARPOL 73/78 Convention (not below 60 °C);
- Asphaltene content from 6.3% to 10.5% along with non-compliance with the requirements to the indicators characterizing the colloidal stability of the fuel (toluene equivalent, xylene equivalent, peptization number) indicates that the use of heavy pyrolysis tar as marine fuel may lead to increased deposits on the heated surfaces of crowd systems, as well as low fuel stability on water content (watering) and formation of low temperature deposits in tanks (ballast residue). Due to the high content of asphaltenes, the product has a high carbon residue (12–16%).

When heavy pyrolysis tar is used in the composition of marine fuel, it is necessary to select the formulation by assessing the compatibility of the components. In order to improve aggregative properties of VLSFO with this component, it is necessary to include high aromatic compounds—for example, catalytic cracking and coking gas oils. Additionally, according to the safety data sheet, this product belongs to the third hazard class (instead of the fourth, according to GOST 10585 requirements).

Based on the above, the use of heavy pyrolysis tar as a component of VLSFO requires careful selection of the other components of the composition to improve fuel stability and to reduce its density below that of water. The component has no sulfur content limits for use as VLSFO.

*3.4. Waste Lubricating Oils*

There are known methods of the utilization of waste motor oils accumulated in appropriate collection points and recycling enterprises as an additive to VLSFO [45,46]. The quality of the collected mixture depends on predominance of oils of these or those

groups in its composition. However, they have common features: the presence of metal ash compounds, increased toxicity, a dark, almost black color and the tendency to emulsify water. The increased ash content of waste motor oils is directly related to the use of additives containing calcium, zinc or phosphorus. The metal content, however, can vary widely in different "batches," as can the types of metals.

Article [47] deals with the joint hydrorefining of waste motor oils with vacuum gas oil and UCO (waste cooking oil) in order to obtain liquid cracking products. Free water and mechanical impurities are first sedimented and then the emulsified water and light fractions are evaporated by heating to 120 °C under atmospheric pressure, reducing the water content to almost zero. Article [48] compared fresh and waste oil by their metal contents and noted increased iron and calcium content. Further use of waste oil is implied in terms of obtaining caloric fuel gas by pyrolysis and thus circumventing, among other things, the problem of its contamination. Processing into fuel gas is also considered in article [49]. The problem of removing ash elements on the example of zinc and phosphorus is also touched upon by other scientists [50], which is also done by pyrolysis and is quite effective: the product contains 4 ppm of phosphorus and 1 ppm of zinc (the "before" concentrations are 647 ppm and 686 ppm, respectively). A summary analysis of the composition of waste motor oils and some of their properties is given in Figure 2.

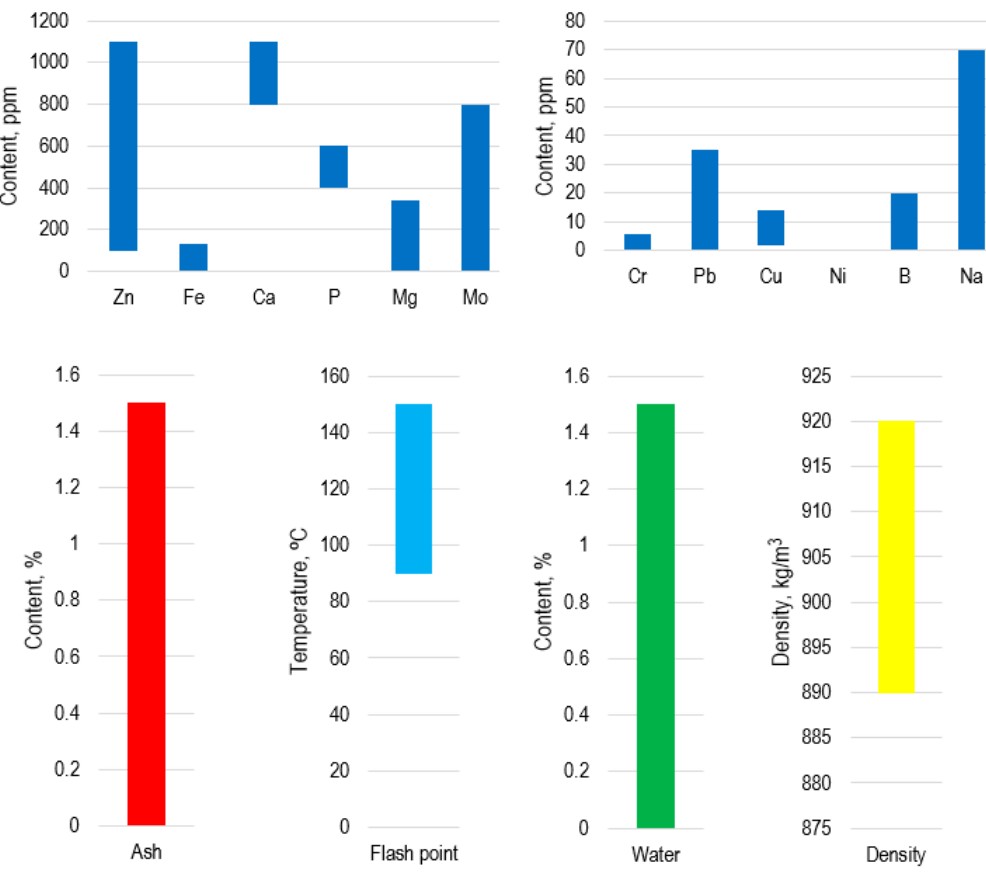

**Figure 2.** Composition and properties of waste motor oils [47–53].

In order to analyze the possibility of the application of waste oils as a component of VLSFO, a sample of a waste oil mixture after its filtration and sedimentation with the following quality indicators was selected and examined by physical and chemical indices in comparison with RMG-380 brand (Table 6). ISO 8217 does not regulate the content of waste motor oil in marine fuels, but it does regulate the calcium and zinc or calcium and phosphorus contents of such fuels.

**Table 6.** Test results of the waste oil mixture.

| Property | Method | RMG 380 | Result |
|---|---|---|---|
| Kinematic viscosity at 100 °C, mm$^2$/s | ASTM D 445 | - | 9.16 |
| Kinematic viscosity at 50 °C, mm$^2$/s | ASTM D 445 | Max 380 | 34.21 |
| Ash content, % | ASTM D 482 | Max 0.1 | 2.851 |
| Water content, % | ASTM D 95 | Max 0.5 | 7.23 |
| Sulfur content, % | ASTM D 2622 | Max 0.5 | 0.312 |
| Hydrogen sulfide content, % | IP 570 | Max 2 ppm | Max 0.5 |
| Flash point in closed cup, °C | ASTM D 92 | Min 60 | 61 |
| Pour point, °C | ISO 3016 | Max minus 30 | Below minus 25 |
| Carbon residue, % | ISO 10370 | - | 3.76 |
| Density at 15 °C, kg/m$^3$ | ISO 12185 | Max 991 | 877.6 |
| Metal content<br>Ca and Zn, ppm<br>Ca and P, ppm | IP 501, IP 470, IP 500 | Ca > 30 Zn > 15 or<br>Ca > 30 P > 15 | - |

The test results of a sample of waste oils show some disadvantages of this product as a fuel component for ship power plants, which is expressed in:

- Increased ash content due to high content of organometallic additives, combustion products (fuel, oil), metal chips formed as a result of running and lapping engine parts, which means insufficient purification of the sample from metals;
- High water content, which is also due to insufficient evaporation of water, as it is known from the sources above that it is possible to obtain oil with a water content of 0–1.5%.

Thus, to involve waste oils in the composition of marine fuels, it is worth paying attention to the process of their preparation and ash content, additionally investigating the contents of calcium, zinc and phosphorus. The limit of such an oil introduction into fuel will be determined by its initial quality; nevertheless, apart from the mentioned indicators, there are no limitations.

### 3.5. The Toxicity for People and Animals

Since the key problem of the components indicated above is the hazard class, it is worth considering separately their toxicity to living organisms. An indicator of toxicity is the LD$_{50}$, which is the average dose of a substance that causes the death of half the members of the test group of animals (rats or rabbits, for example). For all components, except for phenol tar, data are taken from safety data sheets. For tar, in the sample tested in this work and in [37], it is calculated by the average content of the components (phenol—7%, acetophenone—15%, a -methyl styrene dimers—9%, cumenyl phenol—45%, dimethyl phenyl carbinol and others—25%). The toxicity of the investigated wastes is shown in Table 7.

**Table 7.** Toxicity of the investigated wastes of oil refining and petrochemistry.

| Component | LD$_{50}$ (mg/kg)-Oral-Rat | LD$_{50}$ (mg/kg)-Dermal-Rabbit |
|---|---|---|
| **Phenol tar** | **2100** | **3100** |
| Phenol [54] | 317 | 630 |
| Dimethyl phenyl ethyl carbinol [55] | 3260 | 2500 |
| Acetophenone [56] | 815 | 3300 |
| Alpha methyl styrene dimers [57] | 4900 | 14,560 |
| Cumenyl phenol [58] | 1770 | >2000 |
| **VAT distillation residue of butyl alcohols** [59] | **>5000** | **>2000** |
| **2-ethylhexanol (the heavy product of its rectification)** [60] | **3730–4050** | **1970** |
| **Heavy pyrolysis tar** [61] | **2500** | **>1700 (dimethylbenzene)** |
| **Waste lubrication oil** [62] | **>2000** | **>4480** |
| **Conventional fuel oil** [63] | **5270** | **2000** |

Comparison of the toxicity of the components with marine fuel shows that the VAT distillation residue of butyl alcohols and the heavy rectification product of 2-ethylhexanol have approximately the same lethal doses. Phenol tar and heavy pyrolysis tar have a lower lethal dose due to them containing hazardous compounds such as phenol and acetophenone (phenol tar) and benzopyrene (heavy pyrolysis tar).

*3.6. Elaboration of Requirements for Low-Sulfur Marine Fuels*

The involvement of wastes that do not contain sulfur brings up a problem: to what extent can low-sulfur fuels ensure the performance of engines and fuel equipment? Accordingly, the next issue considered in the paper is the study of properties of low-sulfur marine fuels with sulfur content up to 0.5%, which is shown by the example of oil products from two manufacturers with the following quality (Table 2) using a depressor-dispersant additive.

A series of compositions of laboratory samples of marine fuel were developed, guided by the following criteria and approaches:

1. Providing the necessary range of values of kinematic viscosity of fuel. On the one hand, viscosity should be close to the upper limit (no higher than 36.2 mm$^2$/s at 50 °C) to minimize the composition of the fuel distillate components with high production costs; on the other hand, it should not be below 8.9 mm$^2$/s at 50 °C—to ensure the proper operation of fuel automatics. The viscosity of petroleum product mixtures was predicted on the basis of calculations by the method in [64].
2. The target value of the mass fraction of sulfur (not more than 0.5 % wt. for marine fuel meeting the requirements of MARPOL 73/78).
3. Increased requirements for the specific heat of combustion of the marine fuel (at least 42,454 kJ/kg)—the main combustibility characteristic of the fuel.
4. Ensuring the required level of the low-temperature pouring ability of the fuel (pour point, not higher than minus 5 °C) and its stability, which is determined by the selection of an additive in its economically reasonable concentrations.
5. The provision of other quality indicators specified in ISO 8217, as well as indicators of fuel performance properties.
6. Ensuring the level of fuel biostability at a level no worse than petroleum fuels of conventional composition.

Table 8 shows the results of tests of bunker fuel oil made from the components of Refinery № 1, Refinery № 2 and sulfur fuel oil of Refinery № 3. Samples VLSFO № 2 and № 3 were produced from industrial generated components of Refinery № 1 and Refinery № 2. Samples of VLSFO № 4, 5, 6, 7, 8 and 9 were produced from a mixture of

the components of the two mentioned enterprises in order to evaluate the possibility of improving the protective properties of the components' mixture produced by Refinery № 1 by the introduction of a hydrotreated vacuum-distillate fraction with higher sulfur content.

**Table 8.** Quality indicators of laboratory samples of sulfurous (№ 1) and low-sulfur (№ 2–9) marine fuel.

| Property | Requiments | Samples | | | | | | | | |
|---|---|---|---|---|---|---|---|---|---|---|
| | | № 1 | № 2 | № 3 | № 4 | № 5 | № 6 | № 7 | № 8 | № 9 |
| Density at 15 °C according to ISO 3675, kg/m³ | Max 958.3 | 927.7 | 884.7 | 859.9 | 885.60 | 880.80 | 885.6 | 882.9 | 872.5 | - |
| Flash point in closed cup according to ISO 2719, °C | Min 80 | 82.0 | 81.5 | 91.5 | 80.5 | 83.0 | 81.0 | - | - | 81.5 |
| Water content по ISO 3733, % | Max 0.3 | отс. | отс. | отс | отс. | отс | отс | отс | отс | отс |
| Kinematic viscosity at 50 °C according to ISO 3104, mm²/s | Max 36,2 | 32.35 | 11.1 | 10.0 | 10.5 | 11.2 | 8.54 | 9.15 | 10.06 | 6.7 |
| Pour point according to ISO 3016, °C | Max 5 | Below minus 13 | Minus 6 | Below minus 11 | Minus 6 | Minus 6 | Minus 6 | Above 5 | 3 | Minus 14 |
| Demulsibility, amount of separated water, % vol. | Min 10 | 70 | 99 | 99.5 | 99 | 99.5 | 99- | 99 | 99- | 99 |
| Optical density of sediment solution, units | Max 0.560 | 0.555 | 0.020 | 0.020 | 0.024 | 0.024 | 0.024 | 0.024 | 0.024 | 0.024 |
| Specific heat of combustion in terms of dry fuel according to GOST 21261, kJ/kg | Min 42,454 | 41,460 | 42,500 | 42,940 | - | - | - | - | - | - |
| Sulfur content по ISO 8754, % | Max 1.50 | 1.495 | 0.00125 | 0.254 | 0.0276 | 0.0174 | 0.0609 | 0.0595 | 0.0556 | 0.0996 |
| Carbon residue по ISO 10370, % масс | Max 6.0 | 5.07 | absence | absence | absence | absence | absence | absence | absence | absence |
| Stratification, the amount of sediment, % wt., at temperature, °C: 20 60 | Max 10 from 0 to 0.7 | 4.71 0.05 | 7.2 0 | 0.09 0 | - | - | - | - | - | - |
| Corrosivity (loss of metal), g/m² | Max 10 | 0.16 | 25.36 | 4.76 | 14.14 | 24.19 | 24.55 | 25.75 | 26.25 | 16.26 |

Analysis of the data (Table 8) showed that all the developed compositions of low-sulfur fuel (samples № 2–9) differ from the sulfur fuel № 1 by significantly lower coking (susceptibility to high-temperature deposits), optical density of sediment solution (susceptibility to low temperature deposits), demulsibility (stability during storage), mass fraction of sulfur (compatibility with materials, corrosivity, environmental friendliness) and specific heat of combustion (combustibility).

Samples of low-sulfur fuel have considerably higher combustion efficiency indicators, as their composition excludes the formation of unburnable residue, which can reach 6% wt.% in sulfur fuel oil. A higher stability of VLSFO properties ensures the supply of cleaner fuel to the boilers, free from water and impurities, which together with increased combustion heat allows the increasing of the efficiency of the ship power plant.

According to results of the biostability assessment of experimental laboratory samples of VLSFO performed according to the GOST 9.023 procedure, the following results were obtained:

- No signs of mold fungi and bacteria development were found in experimental laboratory samples № 1–№ 9, protected by a biocide additive on the basis of beta-nitrostyrene (0.05 % wt.);
- The signs of growth of mold fungi and bacteria in samples № 1–№ 9 without biocide additive are identical.

The results of evaluation of biostability of experimental laboratory samples of VLSFO with addition of phenol tar and VAT distillation residue of butyl alcohols showed practically no signs of mold fungi and bacteria development, which allows the assumption that phenol tar and the VAT distillation residue of butyl alcohols have biocidal properties. However, this assumption is subject to further study.

A disadvantage of low-sulfur bunker fuel oil is the non-compliance of most samples (№ 2, 4–9) with the established requirements for protective property, i.e., the ability to

prevent the electrochemical corrosion of storage and pumping equipment elements due to the neutralization of the electrochemical potential of metal surfaces with the help of sorption-active polar compounds. This is partly due to the reduced sulfur content (up to 0.001 % wt.), which is a consequence of deep hydrotreatment processes and the removal of the main mass of heteroatomic compounds. The above is confirmed by the correlation between the sulfur content and protective properties of the laboratory samples and nine sampled pilot batches of marine fuel oil investigated in the present work. The dependence on the corrosivity of sulfur content is shown in Figure 3.

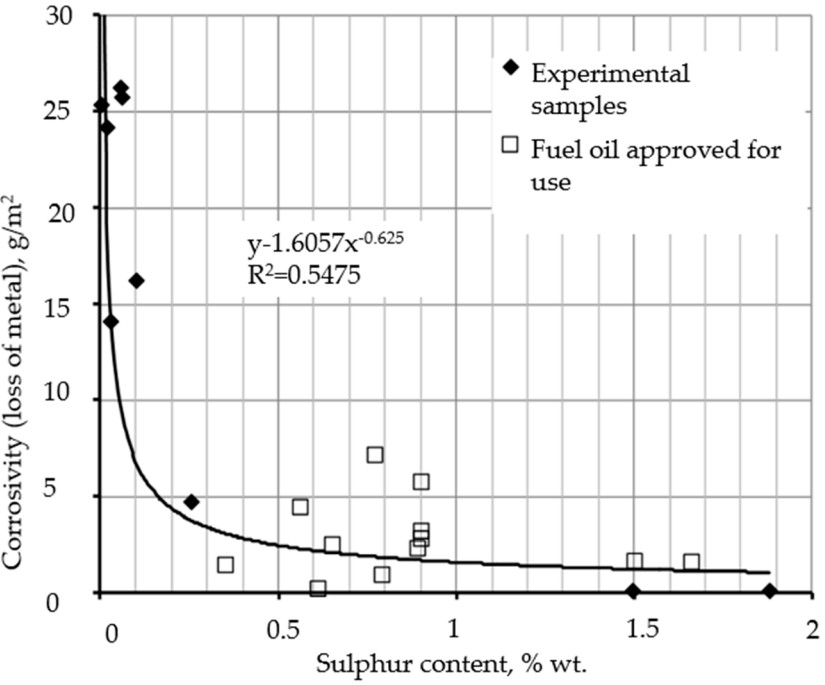

**Figure 3.** Correlation of metal losses during alternate contacting of fuel with water (protective property) with the mass fraction of sulfur.

The research of the change of protective ability at the change of sulfur content in laboratory samples № 4–9 has shown the absence of a direct correlation of the protective properties and depth of hydrotreatment in the range from 0.001 % wt. to 0.1 % wt.

The available data on the results of long-term (up to 150 h) engine tests in power plants of similar in composition low-sulfur fuel have shown that its use will provide (compared with sulfur fuel oil) a reduction in the smoke of exhaust gases and fuel consumption and reduce the negative impact of exhaust gases on the radar, navigation, optical and other equipment of the upper deck, and reduce the radiation level of flue gases.

Analysis of VLSFO application in shipboard machinery for the last 50 years showed that its quality in fuel tanks and at supply to power units in most cases differs from standard requirements by such indicators as water content and mechanical impurities, which is connected with the low stability of VLSFO during storage and negatively influences the above-mentioned performance characteristics of the units.

Taking into account the stated above, the values of changes (improvements) of the power units' performance characteristics when using VLSFO will be higher than similar changes established during engine tests due to the more unfavorable conditions of fuel oil operation.

Based on the studies performed, the requirements for VLSFO quality parameters were elaborated. In terms of physical–chemical and operational properties, VLSFO must meet the requirements given in the Table 9.

**Table 9.** VLSFO quality indicators.

| Property | Requirements | Method |
|---|---|---|
| 1 Kinematic viscosity at 50 °C, mm$^2$/s, Max: Min | 36.20 8.90 [1] | ISO 3104 |
| 2 Ash, %, Max | 0.05 | ISO 6245 |
| 3 Mechanical impurities content %, Max | 0.10 | GOST 6370 |
| 4 Water content, %, Max | 0.3 | ISO 3733 |
| 5 Content of water-soluble acids and alkalis | Absence | GOST 6307 with addition according to 7.5 GOST 10585 |
| 6 Sulfur content, %, Max | 0.50 [2] | ISO 8754 |
| 7 Carbon residue, %, Max | 6.00 | ISO 10370 |
| 8 Hydrogen sulfide content, ppm, Max | 10 | IP 570 |
| 9 Flash point in closed cup, °C, Min | 80 | ISO 2719 |
| 10 Pour point, °C, Max | Minus 5 | ISO 3016 |
| 11 Calorific value (lowest) in terms of dry fuel, kJ/kg, Min | 42,454 [2] | GOST 21261, GOST 34210 |
| 12 Density at 15 °C, kg/m$^3$, Max | 958.3 | ISO 3675, ISO 12185 |
| 13 Optical density of the toluene-acetate solution of the precipitate (ОП$^3_{360}$), Max | 0.560 [3] | According to internal standard |

[1] The requirement allows for stable operation of the fuel system on the VLSFO at the highest possible fuel temperature in the flowing tank. [2] The norm is used to improve the performance of power facilities. [3] The use of the standard prevents the use of VLSFO contaminated with substances that increase the propensity of fuel oil to sediment and form persistent emulsions with water.

## 4. Discussion

One of the obstacles to the introduction of refinery and petrochemical waste studied in this work is their hazard class (2 and 3), which is different from marine fuel (4). In practice, this means that such fuels will require additional tests during their passportization and the concentrations of involvement are severely limited. Also, the quality of such wastes can be further improved, which, however, will increase their cost and possibly compete with hydrotreated components. General characteristics of the possibility of waste involvement in VLSFO are presented in the Table 10.

**Table 10.** Possibility of involving studied wastes in VLSFO.

| Component | Involvement Limit, % | Limiting Characteristics/Property |
|---|---|---|
| Phenol tar | 0–5 | The product contains a small amount of sulfur. The presence of phenol limits its content in marine fuel by lowering the hazard class. Significant content of oxygen-containing compounds and chemical instability. |
| VAT distillation residue of butyl alcohols, heavy rectification product of 2-ethylhexanol | 0–15 | The product contains little sulfur. Limits the hazard class (3). Flash point must pass. |
| Heavy pyrolysis tar | 0–5 | The product is low in sulfur, yet has a high carbon residue, low flash point and high density. To be able to use this component in marine fuel, its density must be reduced to at least 1010 kg/m$^3$ (RMK), but more promising to 991 kg/m$^3$ (RMG). Chemical instability. |
| Waste motor oil | 0–10 | Limiting factor: ash content, and in particular the content of zinc, phosphorus and calcium. |

Further research is needed on the compatibility of the studied components with conventional petroleum components in terms of studying the stability of such systems. The second limitation of the presented approach is the lack of motor bench tests of new

components in the composition of marine fuels, which will allow us to fully evaluate the effect of compounds on the engine.

It has been established that application of VLSFO produced on the basis of residual and distillate fractions of the hydrogenation processes of oil refining provides the stability of its quality during storage, an increase in intervals of necessary maintenance of boiler equipment and an increase in productivity of ship power plants. Reduced sulfur content may lead to the need for additional study of the corrosivity of new components, possibly as marine fuel or the need to use corrosion inhibitors.

## 5. Conclusions

Oil refining and petrochemical wastes are promising components of marine fuels due to the reduced content of sulfur compounds. In addition, their use effectively solves the problem of waste disposal. In this work, components such as phenol tar, VAT DRBA and HPRT, heavy pyrolysis tar and waste lubricating oils were investigated for compliance with the ISO 8217 standards for RMG-380. Compositions with different sulfur contents were studied to determine the weak points of such fuels and to develop requirements for them. The main findings of the research can be summarized as follows:

- Among all the studied components, the most preferable was the use of VAT DRBA and HPRT: their toxicity from animal danger point of view is comparable to that of marine fuel, and the quality fully meets the requirements of ISO 8217. It is advisable to use no more than 15% as a primary analysis; however, an involvement limit should be proven in further studies.
- Phenolic resin and heavy pyrolysis tar contain hazardous compounds that require either additional purification or significant limitations on the content of the component in the fuel.
- The limitation factor for used lubricating oils is the presence of metals in the composition, since all other characteristics, when the oils are given proper preparation, are within the ISO 8217 regulations.
- Application of marine fuel with sulfur content less than 0.1 % wt. can be difficult due to the deterioration of the corrosion properties, which requires additional adjustment of the composition or the use of additives.

**Author Contributions:** Conceptualization, E.A.S. and V.D.S.; Methodology, M.A.E., U.A.M., N.A.K. and M.M.L.; Formal analysis, M.I.F., A.A.S. and A.E.M.; Investigation, D.V.N., M.I.F., A.A.S. and E.O.T.; Writing—original draft, U.A.M., A.E.M. and M.Y.R.; Writing—review & editing, E.A.S., M.A.E., N.A.K., V.D.S. and M.M.L.; Visualization, M.Y.R.; Supervision, D.V.N., M.A.E. and V.M.K.; Project administration, E.O.T. All authors have read and agreed to the published version of the manuscript.

**Funding:** This research received no external funding.

**Institutional Review Board Statement:** Not applicable.

**Informed Consent Statement:** Not applicable.

**Data Availability Statement:** Not applicable.

**Conflicts of Interest:** The authors declare no conflict of interest.

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
