# Peer review of "New Prospects of Waste Involvement in Marine Fuel Oil: Evolution of Composition and Requirements for Fuel with Sulfur Content up to 0.5%"

_jmse, doi:10.3390/jmse11071460_

Round 1
Reviewer 1 Report
This paper considers possibility of addinfg oil refining wastes and petro-16 chemical by-products in Marine Fuel Oil, with respect to usual goals to reduce emission footprint of marine sector. As such it is interesting and worthy of consideration for publication. However, some substantial modifications are needed prior its final consideration for publishing, mainly related to the writing style.
- please avoid statements "on the figure" - you should write "in the Figure..." mentioning the exact number of the figure - both of the figures are not directly mentioned in the text
- the Introduction should be completely reorganized - please write the state-of-the art (avoid lumping references), identify research gap and other references (these 3 might be very helpful and I think they should be used to strengthen the reference list: https://www.sciencedirect.com/science/article/abs/pii/S0306261921016883, https://www.sciencedirect.com/science/article/abs/pii/S1364032121006493, https://www.sciencedirect.com/science/article/pii/S0301421522000945) and clearly elaborate how this paper fills into that gap
- after that, introduce methods and results, and within the discussion section, please state weaknesses and limitations of your approach
- finally, ADD CONCLUSION - this is very important issue, and make sure that your concluding remarks are useful and that they origin from your result
The paper should be reviewed again.
More or less this is okay, but will be checked in details after final review.
Author Response
- Corrected
- The Introduction was expanded and knowledge gap was highlighted
- Corrected
- Conclusion was also added
Reviewer 2 Report
The article "New Prospects of Waste Involvement in Marine Fuel Oil: Evolution of Composition and Requirements for Fuel with Sulfur Content up to 0.5%" is interesting for journal readers and has good scientific content. The use of energy alternatives in combustion engines will be one of the great challenges in the coming years. But for its publication, authors are recommended to review the following aspects.
1. It is recommended not to use capital letters in the middle of the text if they will not be acronyms. ex. Marine Fuel Oil.
2. The use of acronyms must be described in the text.
3. Do not use acronyms in keywords.
4. Table 1 should be associated with the bibliographic references.
5. Section 2.2. Methods, should be explained in more detail instead of dedicating a simple sentence to each method.
6. Can the authors elaborate on the toxicity for people and animals and the implications of the use of the different compounds?
7. There is no reference to ash production and exhaust gas after-treatment systems (for example, exhaust with water) and how these can affect the environment and marine life.
8. A clearer conclusions section is missing and not indicate it as a discussion.
9. The authors should refer to the performance of the fuels since most of the time marine engines work in stationary modes.
Author Response
- Corrected
- Corrected
- Corrected
- Corrected
- The section has been expanded to detail the performance of corrosiveness and biostability tests, as well as the calculation of LD50 for mixtures.
- A separate subsection 3.5 has been added to compare the toxicity of components with marine fuel.
- Added to Introduction
- Conclusion was also added
- The article studied the properties of components for compliance with the requirements of ISO 8217, the influence of the engine operating mode was not evaluated, however, it is a promising direction for the development of this topic.
Round 2
Reviewer 1 Report
The paper seems to be fine, and can be accepted now. Please note that just after Equation 1, both W1 and W2 should be defined. In this way there is a typo - W2 is missing. Also, in Ref. 15 the last author is missing - please, check!
The paper can be accepted and above issues can be resolved in proofreading stage.